# Peptides from the Sea Anemone *Metridium senile* with Modified Inhibitor Cystine Knot (ICK) Fold Inhibit Nicotinic Acetylcholine Receptors

**DOI:** 10.3390/toxins15010028

**Published:** 2022-12-30

**Authors:** Igor E. Kasheverov, Yulia A. Logashina, Fedor D. Kornilov, Vladislav A. Lushpa, Ekaterina E. Maleeva, Yuliya V. Korolkova, Jinpeng Yu, Xiaopeng Zhu, Dongting Zhangsun, Sulan Luo, Klara Stensvåg, Denis S. Kudryavtsev, Konstantin S. Mineev, Yaroslav A. Andreev

**Affiliations:** 1Shemyakin-Ovchinnikov Institute of Bioorganic Chemistry, Russian Academy of Sciences, str. Miklukho-Maklaya 16/10, 117997 Moscow, Russia; 2Institute of Molecular Medicine, Sechenov First Moscow State Medical University, Trubetskaya str. 8, bld. 2, 119991 Moscow, Russia; 3Moscow Institute of Physics and Technology, Institutsky per., 9, 141700 Dolgoprudnyi, Russia; 4Medical School, Guangxi University, Nanning 530004, China; 5Faculty of Biosciences, Fisheries and Economics, Norwegian College of Fishery Science, UiT—The Arctic University of Norway, NO 9037 Tromsø, Norway

**Keywords:** sea anemone venom, *Metridium senile*, ICK fold peptides, nicotinic acetylcholine receptors, NMR structure, electrophysiology, radioligand assay

## Abstract

Nicotinic acetylcholine receptors (nAChRs) play an important role in the functioning of the central and peripheral nervous systems, and other organs of living creatures. There are several subtypes of nAChRs, and almost all of them are considered as pharmacological targets in different pathological states. The crude venom of the sea anemone *Metridium senile* showed the ability to interact with nAChRs. Four novel peptides (Ms11a-1–Ms11a-4) with nAChR binding activity were isolated. These peptides stabilized by three disulfide bridges have no noticeable homology with any known peptides. Ms11a-1–Ms11a-4 showed different binding activity towards the muscle-type nAChR from the *Torpedo californica* ray. The study of functional activity and selectivity for the most potent peptide (Ms11a-3) revealed the highest antagonism towards the heterologous rat α9α10 nAChR compared to the muscle and α7 receptors. Structural NMR analysis of two toxins (Ms11a-2 and Ms11a-3) showed that they belong to a new variant of the inhibitor cystine knot (ICK) fold but have a prolonged loop between the fifth and sixth cysteine residues. Peptides Ms11a-1–Ms11a-4 could represent new pharmacological tools since they have structures different from other known nAChRs inhibitors.

## 1. Introduction

Sea anemones are opportunistic polyphagous predators [1]; they settle in one place and actively move only in certain conditions. Moreover, they have no vision nor a centralized or coordinated nervous system; therefore, they use tentacles armed with venom to catch prey [2]. Although they could explore different ways to obtain nutrition [3], sea anemones utilize toxins for prey capture and defense. Tentacles of sea anemones are armed with stinging cells [4]. The special subcellular organelles (nematocysts) are responsible for the stinging ability of these cells: they contain highly folded tubules that could discharge when in contact with something or somebody [5]. These tubules are full of a venom that consist of a variety of compounds to ensure a powerful effect on preys, concurrent species and predators [6]. A wide range of physiological effects were described for sea anemone toxins including cardiotoxicity, pain, dermatitis, paralysis, swelling and necrosis [7]. Peptides and proteins are the most studied and promising components of these venoms [6]. The cnidarian venom system has developed for millions of years [8], and active peptides exemplify exceptional molecular diversity. At least 17 different molecular scaffolds were reported for cysteine-rich peptides of sea anemones [2].

Cysteine-rich peptides exhibit a wide range of biological activities and are interesting both as tools for studies of different molecular targets (channels, receptors, enzymes) and as probable therapeutic leads [9]. Their effect on different ion channels, mainly voltage-gated sodium and potassium channels, is most well studied. More intriguing is that analyses of different sea anemones’ venoms revealed modulators of acid-sensing ion channels (ASIC) [10,11,12,13,14] and transient receptor potential ion (TRP) channels [15,16,17], antimicrobial compounds [17,18,19] and inhibitors of serine proteases and α-amylase [20,21,22,23].

The nicotinic acetylcholine receptors (nAChRs) are ionotropic ligand-gated receptors involved in neurotransmission and different signal cascades in the central and peripheral nervous system [24,25]. Different venomous animals such as snakes, cone snails and scorpions exploited nAChRs as a target for peptide toxins that ensure quick immobilization and death of the prey [26,27,28]. We studied venom of sea anemone *Metridium senile* and found peptides with the capacity to bind to distinct subtypes of nAChRs as well as to inhibit their function. The NMR study revealed a modified inhibitor cysteine knot (ICK) fold of these peptides, which allows the effective interaction with the muscle and α9 nAChR subtypes.

## 2. Results

### 2.1. Isolation of nAChR-Targeting Peptides and Their Primary Structure Determination

The venom of sea anemone *M. senile* showed significant inhibition of the labeled α-bungarotoxin binding to nAChR. It dose-dependently and competitively inhibited the binding of radioactively labeled α-bungarotoxin ([^125^I]α-Bgt) to both the muscle-type α1β1γδ nAChR (a membrane-bound receptor from the *Torpedo californica* ray electrical organ) and the human α7 receptor subtype (transfected in GH_4_C_1_ cells) (Figure 1). The extent of inhibition towards the muscle-type receptor was noticeably higher than the human α7 nAChR, with 88.5 ± 1.9% inhibition at 0.1 mg/mL and 97.2 ± 0.8% at 1 mg/mL (Figure 1a). For comparison, at the human α7, nAChR 0.1 mg/mL and 1 mg/mL of venom inhibited 15 ± 8% and 74.0 ± 1.2% of specific [^125^I]α-Bgt binding, respectively (Figure 1b); thus, the anticholinergic activity detection was carried out in the next stages only on the muscle-type receptor.

At the first stage, the crude *M. senile* venom was separated on several multicomponent fractions by reverse-phase HPLC, and the activity of the obtained venom portions against *Torpedo* α1β1γδ nAChR was measured (gray shaded area) (Figure 2a) showing the maximum inhibition (94.0 ± 1.5%) (Figure 2b). At the next stage, this active fraction was subjected to further separation on the more hydrophobic column for isolation of individual components (Figure 2c). Three peaks (#1, 3, 9) showed the best effects on the *Torpedo* receptor (Figure 2d), and only one of them (#3) contained a pure peptide (shortly named Ms11a-2) with molecular mass 4653.4 Da (Figure 2c), while the other two were additionally fractionated by reverse-phase HPLC on appropriate columns (Figure 2e,f).

Finally, the active peptide with molecular mass 4594.4 Da (shortly named Ms11-1) was isolated from the multicomponent peak #1 (Figure 2e), and two nAChR-targeting peptides with molecular masses 4918.4 (shortly named Ms11a-3) and 3274.8 Da (shortly named Ms11a-4) were found in fraction #9 (Figure 2f). The average molecular weights of all peptides were estimated by MALDI-TOF mass spectrometry (Appendix A). The N-terminal sequences of isolated peptides were established by Edman degradation, and their full names were proposed according to the nomenclature as in [29] (see Appendix A). Fraction Ms11a-3/Ms11a-4 had a double sequence, but the homology of the Ms11a-3 sequence with Ms11a-1/Ms11a-2 allowed us to split sequences (see Appendix A).

DNA primers for RACE (rapid amplification of cDNA ends) were designed based on N-terminal peptide amino acid sequences, and *M. senile* cDNA was used as a PCR template. The RACE method was used for the primary structure characterization through the analysis of cDNAs coding target peptides. Initially, the 3’-RACE was carried out with corresponding custom-engineered degenerated primers and the universal primer. Hereafter, the 5’-RACE was performed with specially designed reverse primers and the universal primer. PCR products were cloned into a pAL2-TA vector and sequenced. For Ms11a-1–Ms11a-3, a variety of cDNA coding for homologous peptides was found. The N-terminal sequence and detected molecular mass were the criterion for the selection of target peptides’ primary structure. Possible posttranslational modifications were considered in the analyses. The cDNA encoding the new peptides precursors were submitted to the GenBank database (GenBank accession numbers: ON605613, ON605614, ON605615, ON605616), and the deduced sequence of peptides are shown in Figure 3 and Appendix A. Analysis by SignalP 4.1 revealed cleavage sites of signal peptides located between amino acid residues (aa) Ala24 and Gly25 in the precursors of Ms11a-1–Ms11a-3 (Figure 3a). Thus, the precursor polypeptides contain identical signal peptides of 24 aa. The C-terminal dipeptide KK could be post-translationally cleaved in mature peptides Ms11a-1 and Ms11a-2. The deduced amino acid sequence of Ms11a-3 contains C-terminal Gly (Figure 3a) that usually provides C-terminus with amidation (in this case, the native peptide contains amidated Lys42).

The 3′ and 5′ RACE procedures were also used for amplification of Ms11a-4 cDNA. There were also several alternative peptide sequences, and one was chosen by the identity of molecular mass. The precursor consists of a signal peptide (23 aa), propeptide sequence (23 aa) followed by the mature peptide consisting of 32 aa and C-terminal Gly providing amidation Tyr31 (Figure 3b).

The calculated molecular weights of the target peptides, taking into account their posttranslational modifications, practically coincided with that measured by MALDI-TOF (see Appendix A). Alignment of the primary structures of the new peptides with some known toxins of the same structural class 11a showed a low degree of homology (Figure 3c).

### 2.2. Production of Recombinant Peptides and Their Binding and Functional Activity

The *Escherichia coli* expression system was utilized to obtain peptides for the analysis of bioactivity and structural investigations. Peptides Ms11a-1–Ms11a-4 were expressed as a fusion protein with thioredoxin (Trx). Synthetic genes encoding Ms11a-1–Ms11a-4 peptides were constructed by PCR from synthetic oligonucleotides and cloned into the expression vector pET32b (+). Constructions were used to transform *E. coli* BL21 (DE3) cells. Fusion proteins were isolated by metal affinity chromatography and subjected to CNBr cleavage to release the respective recombinant peptides, which were purified with reverse-phase HPLC. The final yields of the target products were estimated to be ~2.4 mg/liter of the cell culture. The recombinant peptides had the same molecular weights as the natural ones, with the consideration of C-terminus amidation absence for Ms11-3a and Ms11a-4. Retention time during the co-injection of natural and recombinant peptides on a reverse-phase column were also identical (data not shown), confirming the proper folding of the recombinant peptides.

All four recombinant peptides were tested for their binding and inhibitory activities with respect to several nAChR subtypes. The ability to compete with [^125^I]α-Bgt for binding to orthosteric sites was studied on muscle-type *T. californica* and human α7 nAChRs (Figure 4).

Concentration-dependent inhibition curves were obtained to calculate the binding parameters (IC_50_ values and Hill slopes) (Table 1). All peptides exhibited a higher affinity towards the muscle-type receptor, Ms11a-3 being the most active with IC_50_ = 255 nM. The lowest activity to both receptor subtypes was detected for peptide Ms11a-4 (Table 1). The peptide Ms11a-3 also showed the most selectivity between α1β1γδ and α7 nAChRs (78 times).

For these reasons, we only investigated the functional activity and selectivity of peptide Ms11a-3 at acetylcholine (ACh)-evoked currents mediated by different nAChR subtypes of some animal species (mouse (m), rat (r) and human (h)) heterologously expressed in *Xenopus laevis* oocytes (Figure 5, Table 2).

Peptide Ms11a-3 was most potent at inhibiting rα9α10 nAChRs (IC_50_ = 202 nM), followed with mα1β1δε (IC_50_ = 1215 nM). The lowest potency of peptide Ms11a-3 was revealed towards α7 and heteromeric nAChRs. The potency of Ms11a-3 at human and rat α7 receptors in electrophysiological tests was in a close range, IC_50_ 8.9 vs. 4.8 μM, respectively, but the potency was less on the human receptor. It can be assumed that the close affinity values obtained from radioligand and electrophysiological tests (IC_50_ 19.8 vs. 8.9 μM; see Table 1 and Table 2) are indirect evidence of blocking the human α7 nAChR function mediated by the interaction of the toxin with the receptor orthosteric binding site. Under this assumption, a more significant difference in affinity and inhibition potency found for the fish and mammalian muscle receptor subtype (IC_50_ 255 vs. 1215 nM, Table 1 and Table 2) calculated from radioligand and electrophysiological measurements, respectively) may reflect not only the specific characters of two different methods (binding/function) but also indicate a certain role of species- or γ/ε subunit-specificities for peptide Ms11a-3.

Peptides (Ms11a-1–Ms11a-4) at 10 µM did not show any activity on rat TRPV1, human TRPV3, rat TRPA1 channels, the human ghrelin receptor, human neurotensin receptor 1 stably expressed in CHO cells in Ca influx assay as well as no effect detected on rat ASIC1a and ASIC3 channels in electrophysiological measurements of *X. laevis* oocytes injected with cRNA coding for these channels (data not shown).

### 2.3. NMR Study of Recombinant Sea Anemone Peptides

The spatial structures of two recombinant peptides Ms11a-2 and Ms11a-3 were solved in water solution by NMR spectroscopy. Our choice of these two toxins for structural studies was due to the greatest difference in their affinity to the muscle-type nAChR (Table 1) with the smallest difference between their amino acid sequences (Figure 3c). Structures were calculated using the following experimental data: torsion angle restraints, upper and lower NOE-based distance restraints, hydrogen and disulfide bond restraints. The statistics of input data and obtained sets of NMR structures are provided in Table 3.

In both cases, the obtained structures converge and are defined well by the experimental data as follows from the low CYANA target function, low RMSD value for backbone atoms and insignificant restraint violations.

Based on the obtained data, the folds of the two peptides are highly similar (Figure 6). Ms11a-2 forms a 3-strand antiparallel β-sheet (strands H8-C9, L20-R23 and W35-R39). There are also two large loops that include four β-turns (N4-S7, II type; Y10-H13, IV; C17-L20, II; N29-G32, IV) and one γ-turn (G32-L34). The conformation is stabilized by 3 disulfides bonds (C2-C17, C9-C22, C16-C37) and 10 hydrogen bonds, according to the hydrogen-deuterium exchange rates (Figure 6a).

Ms11a-3 as well forms a 3-strand antiparallel β-sheet (strands Y8-C9, L20-C22 and W35-R39). Likewise, with Ms11a-2, there are two large loops that include six β-turns (K4-S7, II type; T10-H13, I; R11-R14, VIII; C17-L20, II; P25-G28, IV; G28-R31, IV) and one γ-turn (Y27-I29). The structure is stabilized by 3 disulfides bonds (C2-C17, C9-C22, C16-C37) and 13 hydrogen bonds (Figure 6b).

The surface of both proteins is almost entirely polar (Figure 7). According to the NMR data, in both toxins, the loop (23–36) is partially unstructured and flexible. A comparison of the two structures is shown in Figure 8. The structural scaffold is highly identical; major differences are observed mainly in the region of the second loop, which is partially disordered.

## 3. Discussion

Sea anemones use venom for predation, defense and intraspecific competition [31]. Venoms are composed of different molecules with wide-ranging pharmacological activities [32] which can disrupt physiological processes of other animals [33]. Sea anemones’ venoms have exceptional molecular diversity of polypeptides affecting a variety of targets [2]. To date, the polypeptide toxins interacting with voltage-gated sodium and potassium, ASIC, TRP and other channels have been characterized [2]. Surprisingly, heretofore, none of sea anemones’ peptides was properly characterized as a ligand of nicotinic acetylcholine receptors. Peptides of *H. magnifica* were reported partially to inhibit [^125^I]-αBgt binding to muscle-type *T. californica* at a 20 μM concentration and human α7 nAChR at 40 μM. These peptides failed to inhibit α1β1δε nAChR expressed in *X. laevis* oocytes in electrophysiological experiments. Only one of them showed a potentiating effect on the α7 receptor, but this effect was not properly characterized [14].

nAChRs are the common target of many toxins from various venomous animals. Among them, snakes and *Conus* mollusks are the most famous which produce a huge number of peptides and polypeptides inhibiting nAChRs that predetermine the efficacy of their venoms for victim paralysis (see, for example, reviews [26,27,34]). Recently, an effective interaction with distinct nAChR subtypes was revealed also for some known toxins of scorpions and spiders [28,35,36].

In this work, anticholinergic activity was found in the crude sea anemone *M. senile* venom (Figure 1). The multi-stage purification and identification of active fractions (Figure 2) eventually led to the discovery of four new polypeptides. Their primary structure was established by a combination of protein sequencing, mass spectrometry and cDNA cloning methods (Figure 3 and Appendix A and Appendix A). Recombinant analogues Ms11a-1–Ms11a-4 were produced for adequate analysis of their activity.

Their binding activity showed obvious specificity to the muscle-type nAChR from the electrical organ of the *T. californica* ray, the greatest affinity being by the Ms11a-3 peptide (Table 1, Figure 4). It should be noted that this particular peptide of the four discovered contains the maximum amount of positively charged amino acid residues (11), while the peptide Ms11a-4 with the minimum number of arginines and lysines (4) was significantly less active towards the muscle-type receptor. As was previously reported, the introduction of positively charged amino acid residues into α-conotoxin peptides, targeting muscle nAChR, leads to a noticeable increase in their affinity [37].

The functional activity of the most potent Ms11a-3 towards a large set of different nAChR subtypes from different animal species revealed its antagonism not only to muscle receptors, but also to the rat α9α10 nAChR (IC_50_ = 202 nM) (Table 2, Figure 5). A similar specificity was demonstrated previously for such arginine-rich molecules as α-conotoxin RgIA [38] and αO-conotoxin GeXIVA [39]. It can be assumed that the revealed affinity of the novel toxins from anemones (excluding Ms11a-4) to the nAChRs is determined essentially by the presence of positively charged amino acid residues in their structure.

However, apart from the high basicity, there is no other similarity of the new anemone peptides with those conotoxins. The cysteine pattern (C1-C4, C2-C5, C3-C6) indicates a possible similarity with the inhibitor cystine knot (ICK) fold that is widely adopted by spiders. Analysis of the spatial structure similarity for one of the novel anemone toxins (Ms11a-3) using the PDBeFold tool [40] showed that this peptide forms the classical ICK fold characteristic of spider voltage-gated ion channel inhibitors (Appendix A). However, no noticeable homology in the amino acid sequence was revealed. Besides, an additional difference is the presence of a prolonged loop between the fifth and sixth cysteines in the novel anemone toxins Ms11a-1–Ms11a-3, which is typically shorter in other known ICK peptides. It is known that some of these peptides are positive allosteric modulators of incest nAChRs (x-hexatoxin-Hv1a, j-hexatoxin-Hv1c and x/j-hexatoxin-Hv1h). Additionally, j-hexatoxin-Hv1c can reverse desensitization of cockroach neurons to nicotine but do not affect human α3-containing and α7 nAChRs [35,41]. The residues between the fifth and sixth cysteines were shown to play an important role in the binding of these spider peptides to nAChRs of insects [35]. The fact that the new anemone peptide Ms11a-4 also has a rather short loop C5-C6 and showed low affinity towards both fish muscle and human neuronal α7 nAChRs may indicate that the prolongation of this loop in Ms11a-1–Ms11a-3 toxins is an important adaptation of the ICK fold for the effective binding of nAChRs of vertebrates.

It should be noted that several peptides with the ICK fold were previously found in sea anemones [2,13,42]. However, even in this case, the less active peptide Ms11a-4 is the closest to these peptides—pi-phymatoxin Pcf1a from *Phymanthus crucifer* [12] and acatoxin 1 from *Anthopleura cascaia* (PDB ID: 6NK9) (Figure 3c). The first of them is able to inhibit acid-sensing ion channels (ASIC) (IC_50_ 100 nM) and voltage-gated potassium channels (K_v_) (IC_50_ 3.5 µM) in rat dorsal root ganglia [13]. Acatoxin 1 is a homologous peptide with a NMR-determined 3D structure (PDB ID: 6NK9), but no specific functions were reported (PDB ID: 6NK9). Another ICK fold peptide BcsTx3 from sea anemone *Bunodosoma caissarum* inhibits voltage-gated potassium channels, and crosslinked by four disulfide bonds, three of which likely form a classic ICK framework, while an additional disulfide bridge most probably stabilizes loop 3 [2,42].

The three disulfide bond patterns in Ms11 peptides are similar to the cysteine connectivity pattern of tachystatin A, a peptide from the Japanese horseshoe crab (*Tachypleus tridentatus*). This peptide however is a chitin binding peptide and shares structural similarities to spider toxins and mammalian defensins. Moreover, cysteine-knotted nemertide α peptides share the same cysteine disulfide pattern and binding to voltage-gated sodium channels with high affinity and selectivity [43,44].

In the course of this study, the elucidation of the NMR spatial structure of two new toxins from sea anemones—Ms11a-2 (Table 3, Appendix A) and Ms11a-3 (Table 3, Appendix A) —confirmed their belonging to the ICK fold peptides with the main difference in the length of the C5–C6 loop. The structures of both peptides are very similar to each other except partially disordered in the NMR structure C5–C6 loop (amino acid residues R23-R36) (Figure 6, Figure 7 and Figure 8); that most probably determines the difference between their activity (Table 1). It should be noted that the backbones of the first 20–25 and last 5–8 amino acid residues of Ms11a-2 and Ms11a-3 toxins are almost identical to all 10 different ICK-peptides shown in Appendix A.

The question arises about the biological significance of the discovered toxins with the modified ICK fold for sea anemones. We do not have any data on other possible targets of these peptides; it remains to assume their necessity as blockers of the nicotine cholinergic system. nAChRs play a significant role in sensory and some motor neurons of invertebrates [45,46]. Involvement of nAChRs in neuromuscular transmission in vertebrates makes these receptors an excellent target for paralysis of these phyla representatives. Therefore, venom components affecting nAChRs could immobilize or mislead the prey or predator from different phyla.

The preferred foods of sea anemone *M. senile* are representatives of zooplankton such as copepods, polychaete larvae, bivalve and gastropod veligers, ascidian larvae, copepod and barnacle nauplii, cyprids [47,48]. Most probably due to multicomponent venom, *M. senile* is subjected to predation only by a few species from different phyla. The sea spider *Pycnogonum littorale* from phylum Arthropoda can suck juices from the anemones [49], which makes it evolutionarily similar to terrestrial ticks. The sea slug *Aeolidia papillosa* from phylum Molluska feeds on *M. senile* [50] and can significantly decrease the population of anemones. Epitonid snail (wentletrap) *Epitonium greenlandicum* was reported to eat *M. senile* [51]. Representatives of subphylum vertebrata that eat *M. senile* are the black bream *Spondyliosoma cantharus* [52] and the winter flounder *Pseudopleuronectes americanus* [53]. The sea slug *Aeolidia papillosa* inhibits the discharge of nematocysts from sea anemone tentacles by special mucus [54], but nothing is known about how other species escape from the venom of sea anemone, but evidently, despite low mobility, *M. senile* is not an easy target for most of the predators. Probably one of the possible reasons could be effective paralyzing toxins affecting cholinergic systems of vertebrates.

Interestingly, *M. senile* is considered safe for human beings, most probably due to a much higher affinity to the fish neuromuscular nAChR than to mammal ones. Evolution of these peptides also provided interesting selectivity to different subtypes of nAChRs—the rather high affinity to α9α10 receptors (that are considered as potential participants in the modulation of pain [38] and hearing loss [55]) and low affinity to α7 and heteromeric nAChRs of vertebrates.

## 4. Conclusions

Novel biologically active peptides Ms11a-1–Ms11a-4 inhibiting nAChRs were isolated from the venom of sea anemone *M. senile*. Structures of the most effective peptides represent the adaptation of the ICK fold with a prolonged loop between Cys5 and Cys6. Peptide Ms11a-3 showed a unique profile of selectivity, effectively blocking the neuronal α9α10 and to a lesser extent muscle nAChR subtypes. Peptides Ms11a-1–Ms11a-4 could represent new pharmacological tools since they are different from other known nAChRs inhibitors’ structures.

## 5. Materials and Methods

*Venom Collection and Fractionation*—Specimens of *M. senile* were collected off the coast of Tromsø, Norway in autumn 2012. The animals were kept in fresh-flowing seawater at 8–12 °C for 1–3 weeks. The venom was collected from living specimens subjecting them to electrical stimulation (160 mA, 10 Hz) and mechanical stimulation by a glass spreading rod. The mucus released by the anemone specimens was rinsed of the animals with a 10 mM ethylenediaminetetraacetic acid (EDTA)/0.1 mM phenylmethylsulfonyl fluoride (PMSF) solution and centrifuged. The samples were desalted by reverse phase solid phase extraction. Sep-Pak C18 Vac cartridge/5000 mg (Waters) was conditioned with acetonitrile (ACN)/0.1% trifluoroacetic acid (TFA) and equilibrated in 0.1% TFA. After sample loading, the column was washed with 0.1% TFA. Elution was performed with a 70% ACN/0.1% TFA solution. The eluate was lyophilized and kept frozen at −20 °C until further HPLC analysis. The HPLC separation was performed on a reverse-phase columns Jupiter C5 (250 × 10 mm, Phenomenex, Torrance, CA, USA), Luna C18 (250 × 10 mm, Phenomenex, Torrance, CA, USA), Synergi Fusion-RP (250 × 3 mm, Phenomenex, Torrance, CA, USA), Vydac (250 × 4.6 mm, Grace, Columbia, USA) using gradients of acetonitrile as shown on Figure 2.

*Mass Spectrometry*—MALDI time-of-flight spectrometry on an Ultraflex TOF-TOF instrument (Burker Daltonik, Bremen, Germany) was used to measure molecular weight. The molecular mass determination was carried out in a linear or reflector positive ion mode. Samples were prepared by using the dried-droplet method with 2.5-dihydroxybenzoic acid (10 mg/mL in 70% ACN solution with 0.1% TFA) as a matrix.

*Amino Acid Sequence Analysis*—Peptides were analyzed by an automated stepwise Edman degradation method on a Procise model 492 protein sequencer (Applied Biosystems, Waltham, CA, USA) according to the manufacturers’ protocol.

*Precursor Determination*—Total RNA of *M. senile* was isolated from the tentacles using TRIsol reagent (Ambion, Austin, TX, USA) according to the manufacturer’s protocol. Reverse transcription of RNA into cDNA was performed using the MINT kit (Evrogen, Moscow, Russia) according to the manufacturer’s recommendations. 3′-RACE was performed to use the universal primer Long1 (GTA ATA CGA CTC ACT ATA GGG CAA GCA GTG GTA TCA ACG CAG AGT) and degenerated primers AMS-d1 (TGC TAT AGA CAR CAY MGN GAR TG), AMS-d2 (GAG TGC TGC CAT GGA YTN GTN TG), AMS-d3 (TGC GCA CAG ACT GGN GGN ACN TG) and AMS-d4 (TGT AGG AAT AWS NAR GAY TGY TG). 5′-RACE was performed with universal primer Long1 and reverse primers AMS-r1 (GTT TGA AGT ATA GTT GTT TAG AGC), AMS-r2 (CCT TTA TTA TAA GCA TCC ATT CA) and AMS-r3 (CTG CAC ACC AAA CCA TGA CAA C). DNA sequencing was performed on the Applied Biosystems 3730 DNA Analyzer.

*Gene Synthesis*—The DNA sequences encoding peptides were constructed by assembly PCR from synthetic oligonucleotides. The Met codon was added to the sequences coding for peptides to provide downstream BrCN cleavage of fusion proteins. The amplified PCR fragments were gel-purified and cloned into the expression vector pET32b+ (Novagen, Pretoria, South Africa).

*Recombinant Peptide Production*—Recombinant peptides were produced as fusion proteins with thioredoxin in *E. coli* BL21(DE3). Expression vectors were transformed into *E. coli* BL21(DE3) cells. Transformed cells were seeded in LB medium with ampicillin (100 μg/mL) and cultivated at 37 °C for 4–6 h up to culture density A600∼0.6–0.8. Then, isopropyl-1-thio-β-D-galactopyranoside was added up to 0.2 mM to induce expression of target fusion proteins. To improve peptides’ folding, the cells were cultivated for 18 h at 25 °C. Then, cells were harvested by centrifugation (5 min at 6000× *g*), re-suspended in a buffer for metal affinity chromatography (400 mM NaCl, 20 mM Tris-HCl, pH 7.5) and disrupted by ultra-sonication. All insoluble particles were removed by centrifugation (15 min at 9000× *g*). The fusion proteins were purified using HisPur NiNTA resin (ThermoScientific, Waltham, CA, USA) using manufacturer protocols. Fusion proteins were cleaved by BrCN with the addition of HCl up to 0.2 M in the dark at room temperature, as described [56]. Target peptides were isolated using HPLC on a reverse-phase column Jupiter C5 (250 × 10 mm). Recombinant peptides were confirmed by N-terminal sequencing and MALDI-TOF mass spectrometry.

*Radioligand Assay*—For competition radioligand binding assays, suspensions of nAChR-rich membranes from the *T. californica* electric organ (0.55 nM α-bungarotoxin-binding sites) in 20 mM Tris–HCl buffer (pH 8.0) containing 1 mg/mL BSA (binding buffer) or hα7 nAChR transfected GH4C1 cells (0.4 nM α-bungarotoxin-binding sites) in binding buffer were incubated for 3.5 h at room temperature with different concentrations of competitors (crude M. senile venom, purified venom fractions, purified toxins, recombinant toxins), followed by an additional 5 min incubation with 0.7–0.9 nM [^125^I]α-Bgt. Nonspecific binding was determined by preliminary incubation of the preparations with 30 μM of α-cobratoxin. The membrane and cell suspensions were applied to glass GF/C filters (Whatman, Maidstone, UK) presoaked in 0.25% polyethylenimine, and unbound radioactivity was removed from the filter by washing (3 × 3 mL) with 20 mM Tris–HCl buffer (pH 8.0) containing 0.1 mg/mL BSA (washing buffer). The bound radioactivity was determined using a Wizard 1470 Automatic Gamma Counter. The binding results were analyzed using the OriginPro 2015 program (OriginLab Corporation, Northampton, MA, USA) fitting to dose–response competition curves where all points were presented as the mean ± standard error (SEM) analyzed from respective triplicates. The calculated IC_50_ and Hill slope values were presented as 95% confidence intervals.

*Electrophysiology*—The adult mature female *Xenopus laevis* was purchased from the Kunming Institute of Zoology, and maintained in our laboratory (17 ± 0.5 °C) for more than 3 months. The frog was anesthetized on ice, and stage V or VI oocytes were separated and digested with collagenase Type I (Sigma-Aldrich, Saint Louis, MO, USA). All procedures were carried out following the guidelines for the care and use of laboratory animals and approved by the Ethics Committee of Guangxi University. Plasmids containing various nAChR subunit genes were kindly provided by the University of Utah. The process for preparation, purification and microinjection of nAChR cRNA was as described previously [57]. The two-electrode voltage-clamp (TEVC) technique was used to identify various nAChR subtypes’ expression in *X. laevis* oocytes, and to evaluate the blocking potency of peptides. The oocytes were perfused in a 50 μL chamber with ND96 solution (96.0 mM NaCl, 2.0 mM KCl, 1.8 mM CaCl2, 1.0 mM MgCl2, 5 mM HEPES, at pH 7.5). The perfusion rate was about 2 mL/min. To apply pulses, acetylcholine (ACh) was applied to the oocyte at 1 min intervals for 2 s. This process was automatically performed by using a distributor valve (ALA Corp., Bethpage, NY, USA). The concentrations of ACh used were 10 μM for α1β1εδ and α9α10, 200 μM for α7 and 100 μM for all others. The inward current evoked by the agonist ACh application was recorded under TEVC amplifier Axon 900A (Molecular Devices, San Jose, CA, USA), at a holding potential of -70 mV. The glass microelectrodes were filled with 3M KCl and had a resistance of 0.5–2 MΩ. The current data were recorded and analyzed using pCLAMP 11.0.3 software (Molecular Devices, San Jose, CA, USA), filtered at 10 Hz and digitized at 100 Hz. Peptide activity was determined by comparing the ACh-evoked current response before and after 5 min incubation with oocytes. The ACh-induced peak current responses in the presence of 5 μL ND96 (supplemented with 0.1 mg/mL bovine serum albumin) were normalized to the three averages as a control. The half-maximal inhibition (IC_50_) values were used to estimate the potency of peptides for various nAChR subtypes. The concentration–response curves were fit to the data by the nonlinear regression equation % response = 100/{1 + ([peptide]/IC_50_)nH} using GraphPad Prism 6 software (GraphPad Software Inc., San Diego, CA, USA). The nH is the Hill coefficient. The IC_50_ values were presented as 95% confidence intervals to calculate peptide potency. Data of the concentration–response curve were shown as mean ± standard error (SEM) and collected from at least three oocytes to ensure reproducibility.

*NMR Study*—All NMR experiments were performed on the Avance III 600 MHz spectrometer (Bruker Biospin, Rheinstetten, Germany) at 30 °C; pH was adjusted to 5.2 in the case of the Ms11a-2 peptide and 4.5 in the case of Ms11a-3 peptide. Totals of 1.4 and 2.0 mg of the peptides were dissolved in 300 uL of H_2_O/D_2_O 95:5 solution. ^1^H, ^13^C and ^15^N assignments were obtained via the standard procedure, based on TOCSY, NOESY (60 ms and 120 ms mixing times), ^1^H^13^C-HSQC and ^1^H^15^N-HSQC spectra [58]. After recording the set of spectra, the sample was liophilyzed and then re-dissolved in pure D_2_O (CIL, Tewksbury, MA, USA) to measure the rate of the proton–deuterium exchange of the amide groups and record additional NOESY (120 ms mixing time), ^1^H^13^C-HSQC and DQF-COSY.

Spatial structures were calculated using the simulated annealing/molecular dynamics protocol as implemented in the CYANA software package version 3.98.5 [59]. Upper interproton distance restraints were obtained by 1/r^6^ calibration of NOESY cross-peaks. Torsion angle restraints and stereospecific assignments were obtained based on the J-couplings and NOE intensities. ^3^J_HNHα_ constants were determined by the line shape analysis of 2D NOESY cross-peaks. Hydrogen bonds were introduced based on the deuterium exchange rates of amide protons on the latest stages of the structure calculation. The disulfide bond connectivities were determined by NMR unambiguously for the Ms11a-3 peptide and were assumed identical in the case of the Ms11a-2 peptide, which did not contradict the experimental data. NMR chemical shifts and coordinates of the Ms11a-2 and Ms11a-3 peptides were deposited to the PDB database under the accession codes 6XYH and 6XYI, respectively.

## Figures and Tables

**Figure 1 toxins-15-00028-f001:**
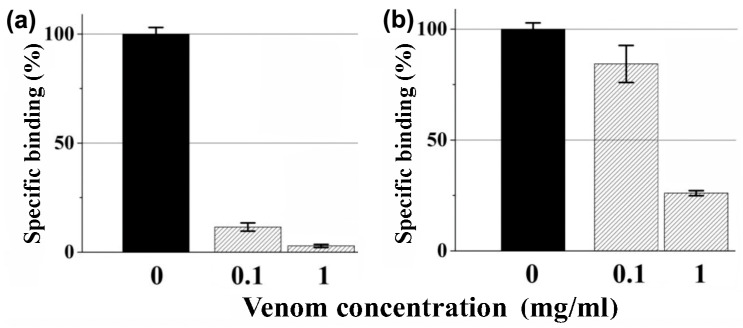
Competition of crude *M. senile* venom at two concentrations with radiolabeled α-Bgt ([^125^I]α-Bgt) for binding to (**a**) *Torpedo californica* α1β1γδ and (**b**) human α7 nAChRs. Each bar is shown as the mean ± standard error of mean (SEM) value of three measurements for each venom concentration in a single experiment. Black bars are corresponding control without the venom.

**Figure 2 toxins-15-00028-f002:**
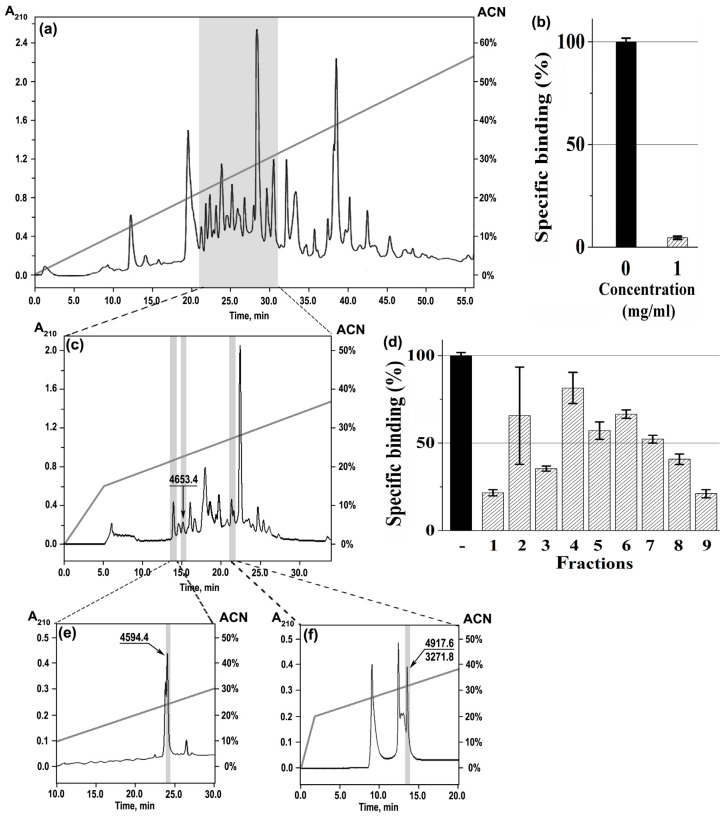
Stages of chromatographic separation of *M. senile* venom by HPLC; absorbance at 210 nm (black line) and gradient line (gray) are shown. (**a**) Fractionation of the crude venom (~5.45 mg) in a linear gradient of acetonitrile (0–60% for 60 min) on a Jupiter C5 (250 × 10 mm) column. (**b**) Competition of the gray shaded fraction with [^125^I]α-Bgt for binding to *T. californica* nAChR (mean ± SEM value from three measurements; black bars correspond to control in the absence of competitor). Concentration fitted to initial amount of the venom. (**c**) Separation of the gray shaded portion from (**a**) in a linear gradient of acetonitrile (0–15% for 5 min, 15–60% for 60 min) on a Luna C18 (250 × 10 mm) column into the *Torpedo* nAChR-targeting fractions characterized with the same radioligand assay conditions The most active fractions—1, 3 and 9—are shown in gray boxes. (**d**) Competition of fractions (final concentration corresponds to 1 mg of the venom) with [^125^I]α-Bgt for binding to *T. californica* nAChR (mean ± SEM value from three measurements; black bars correspond to control in the absence of competitor). (**e**) Isolation of the individual cholinergic ligand in a linear gradient of acetonitrile (0–60% for 60 min) on a Synergi Fusion-RP (250 × 3 mm) column from fraction 1 (**c**). (**f**) Isolation of the active compounds in a linear gradient of acetonitrile (0–20% for 2 min, 20–60% for 40 min) on a Vydac (250 × 4.6 mm) column from fraction 9 (**c**). Target peptides are shown by arrows with corresponding masses (Da).

**Figure 3 toxins-15-00028-f003:**
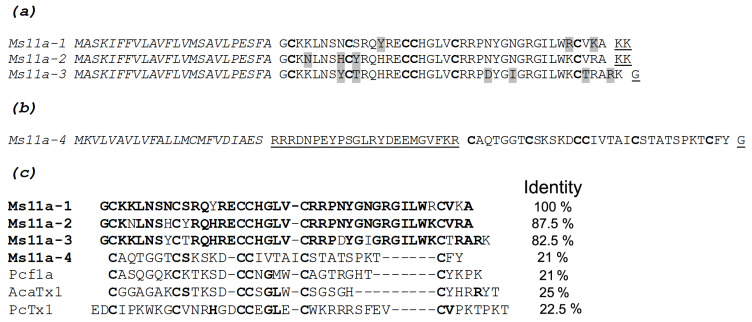
Deduced amino acid sequences of peptides Ms11a-1–Ms11a-3 (**a**) Ms11a-4 (**b**) precursors. The signal peptide is shown in italic; the pro-peptide sequence is underlined; Cys residues are marked in bold; and dissimilar amino acids are highlighted in grey. (**c**) Multiple sequence alignment for four novel peptides from *M. senile* and two other sea anemone toxins from structural class 11a—pi-phymatoxin Pcf1a (UniProtKB ID: C0HJB1) from *Phymanthus crucifer*, a potassium channel inhibitor AcaTx1 from *Anthopleura cascaia* (PDB ID: 6NK9) as well as psalmotoxin PcTx1 (P60514) from tarantula *Psalmopoeus cambridgei*. Identical amino acid residues in peptides are highlighted in bold.

**Figure 4 toxins-15-00028-f004:**
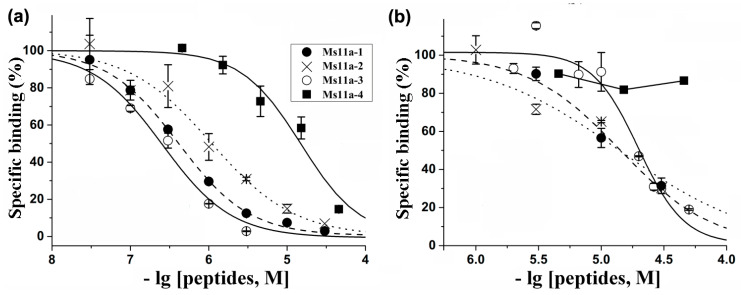
Competition of four recombinant sea anemone peptides with [^125^I]α-Bgt for binding to (**a**) muscle-type α1β1γδ nAChR from *T. californica* ray electric organ and (**b**) human neuronal α7 receptor expressed in GH4C1 cells. Each point presented on the inhibition curves was shown as mean ± SEM analyzed from triplicate. The calculated IC_50_ values and Hill slopes (OriginPro 2015 program) are collected in Table 1.

**Figure 5 toxins-15-00028-f005:**
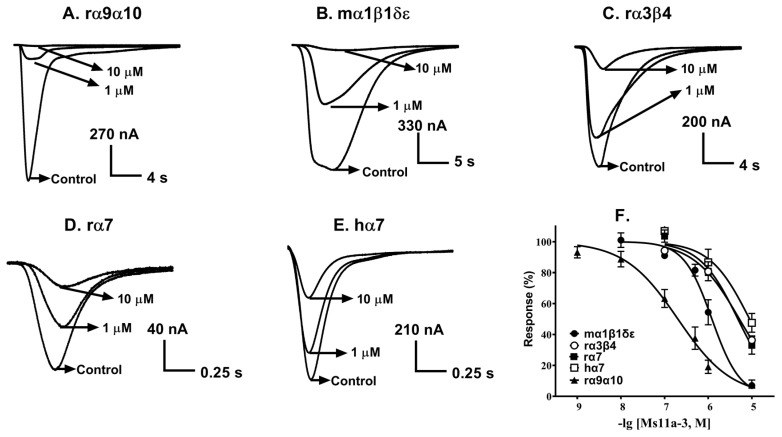
The potency of Ms11a-3 on various nAChR subtypes of some animal species. (**A**–**E**), represent 1 μM and 10 μM concentrations of Ms11a-3 selectively block nAChR subtypes. Control indicates responses to ACh without incubation with Ms11a-3. (**F**). Dose-response curves for the peptide Ms11a-3 at most sensible nAChR subtypes. Each point presented on the dose-response curve was shown as mean ± SEM analyzed from parallel data of 3–6 oocytes. Species’ origin of receptors is rat (r), mouse (m) and human (h). The respective calculated IC_50_ and Hill slope are shown in Table 2.

**Figure 6 toxins-15-00028-f006:**
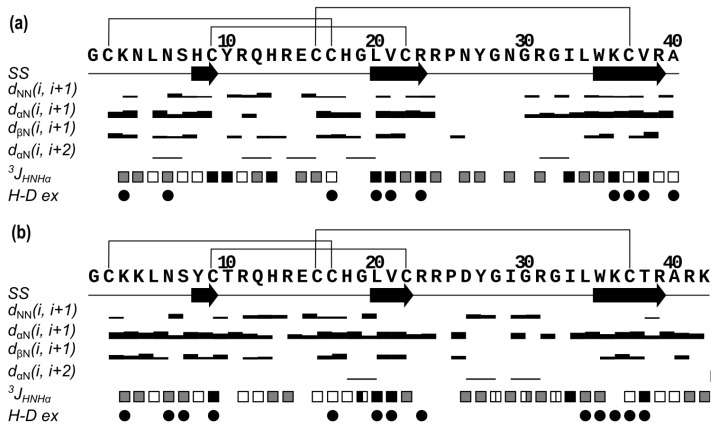
Overview of the NMR data that define the secondary structure of Ms11a-2 (**a**) and Ms11a-3 (**b**). Protein sequence, secondary structure (*SS*), NOE connectivities (d_ij_), ^3^J_HNHα_ couplings and hydrogen-deuterium exchange rate (*H-D ex*) are shown. The arrows indicate strands of the β-strands. Widths of the bars represent the relative intensity of cross-peaks in NOESY spectra. Squares have three colors according to the value of J-coupling: black (large, >8 Hz), grey (medium, 6–8 Hz) and white (small, <6 Hz); for glycines, squares are divided into two rectangles, which correspond to ^3^J_HNHα2_ and ^3^J_HNHα3_ constants. Circles denote the H_N_ protons, with the solvent exchange rates slower than 4 h^−1^. Disulfide bonds are shown as black lines.

**Figure 7 toxins-15-00028-f007:**
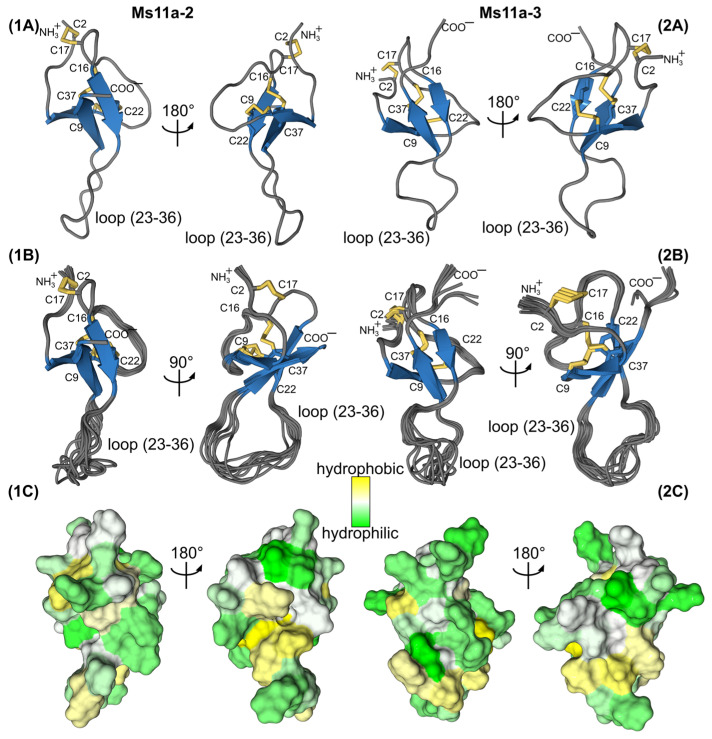
Two-sided view on the Ms11a-2 (**1**) and Ms11a-3 (**2**) spatial structures. (**A**) Representative structures (with the fewest restraint violations) and (**B**) best 10 structures out of initial 100 superimposed over the backbone of β-sheet residues. Disulfide bonds are colored in yellow. (**C**) The contact surface of two molecules is colored according to the hydrophobicity, from yellow (hydrophobic) to green (hydrophilic) using the White and Wimley scale [30].

**Figure 8 toxins-15-00028-f008:**
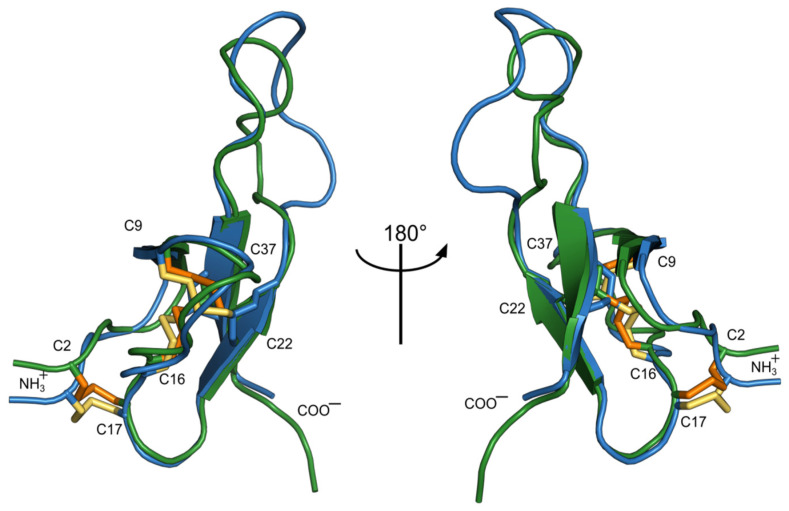
Ms11a-2 (blue, yellow cystines) and Ms11a-3 (green, orange cystines) superimposed over the backbone atoms of the β-sheet residues using PDBeFOLD web-site https://www.ebi.ac.uk/msd-srv/ssm/ (accessed on 28 April 2022).

**Table 1 toxins-15-00028-t001:** Calculated with OriginPro 2015 program from radioligand assay (see the respective inhibition curves in Figure 4) the binding parameters—IC_50_ and Hill slope values—for recombinant sea anemone peptides on muscle-type *T. californica* and neuronal human α7 nAChRs. Data are shown as mean (95% confidence interval).

Peptide	[^125^I]α-Bgt-Competitive Binding Assay			
	*T. californica* α1β1γδ nAChR		Human α7 nAChR	
	IC_50_ (μM)	Hill Slope	IC_50_ (μM)	Hill Slope
Ms11a-1	0.408 (0.395–0.421)	0.96 (0.94–0.99)	14.16 (13.48–14.87)	1.18 (1.11–1.25)
Ms11a-2	1.080 (0.947–12.28)	0.82 (0.75–0.89)	14.13 (12.43–16.07)	0.80 (0.69–0.91)
Ms11a-3	0.256 (0.236–0.277)	0.98 (0.91–1.05)	19.81 (18.44–21.28)	2.37 (2.00–2.75)
Ms11a-4	14.95 (13.46–16.61)	1.14 (1.02–1.26)	>45	-

**Table 2 toxins-15-00028-t002:** IC_50_ and Hill slope values for peptide Ms11a-3 on various nAChR subtypes expressed in *Xenopus* oocytes. Data are shown as mean (95% confidence interval).

nAChR Subtype ^a^	IC_50_ (μM)	Hill Slope
mα1β1δε	1.215 (0.904–1.633)	1.35 (0.71–1.99)
rα2β2	na ^b^	-
rα2β4	na ^b^	-
rα3β2	na ^b^	-
rα3β4	5.173 (4.19–6.388)	0.84 (0.67–1.00)
rα4β2	na ^b^	-
rα4β4	na ^b^	-
rα6/α3β4	na ^b^	-
rα7	4.786 (3.055–7.497)	0.99 (0.59–1.40)
hα7	8.869 (5.218–15.08)	0.95 (0.30–1.61)
rα9α10	0.202 (0.141–0.29)	0.71 (0.52–0.90)

^a^ Species origin of receptors is rat (r), mouse (m) and human (h); ^b^ na = not active (no inhibition observed with 10 μM of Ms11a-3 analyzed on 3–6 oocytes).

**Table 3 toxins-15-00028-t003:** Input data and validation statistics for the best 10 structures of Ms11a-2 and Ms11a-3.

Peptide	Ms11a-2	Ms11a-3
PDB ID	6XYH	6XYI
**Distance and angle restraints**		
Total NOEs	467	558
intraresidual	102	267
interresidual	365	291
sequential (|i-j| = 1)	110	86
medium range (1 < |i-j| ≤ 4)	42	36
long-range (|i-j| > 4)	213	169
Hydrogen bond restraints (upper/lower)	30/30	41/41
S-S bond restraints (upper/lower)	9/9	9/9
J-couplings	31	38
J_HNHα_	31	38
Angles	22	19
χ_1_	19	19
χ_2_	3	0
**Total restraints/per residue**	598/15	715/17
**Statistics of the obtained set of structures**		
CYANA target function	1.26 ± 0.13	2.12 ± 0.19
Restraints violations		
distance (>0.2Å)	2	5
angle (>5°)	1	3
RMSD (Å)		
for all residues		
backbone	0.98 ± 0.2	1.14 ± 0.32
all heavy atoms	1.98 ± 0.3	2.21 ± 0.42
**Ramachandran analysis**		
% residues in most favored regions	66.7	60
% residues in additional allowed regions	33.3	40
% residues in generously allowed regions	0	0
% residues in disallowed regions	0	0

## Data Availability

Not applicable.

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
