# Peer review of "Peptides from the Sea Anemone Metridium senile with Modified Inhibitor Cystine Knot (ICK) Fold Inhibit Nicotinic Acetylcholine Receptors"

_toxins, 2022, doi:10.3390/toxins15010028_

Round 1

Reviewer 1 Report

This paper looks at the functional characterisation of some novel peptides with nAChR activity and a variation on the ICK fold. Function characterisation of any peptide is always important and a contribution to science to extend our knowledge, though should be mindful they may not always be translatable into a bio-application as a pharmacological tool. I recommend minor corrections prior to acceptance.

I have the following specific feedback:

nAChR activity has been isolated for sea anemone peptides see Toxins 202214(10), 697; https://doi.org/10.3390/toxins14100697 modify discussion to reflect this work and include

Throughout the paper there are minor grammar changes to be fixed e.g., in the title “the” should be before sea anemone – this type of error is commonly found throughout the manuscript

Introduction – The ecology for sea anemones is incorrect in the first sentence. They are not predatory – they do not actively hunt their food but are opportunistic feeders. Additionally, their movement is not restricted but they do tend to settle in the one place giving the impression they are sessile, but they are not (coral is an example of sessile). If conditions are unfavourable, sea anemones will most definitely move.

The authors should cite primary research other than relying on reviews.

This paper looks at the functional characterisation of some novel peptides with nAChR activity and a variation on the ICK fold. Function characterisation of any peptide is always important and a contribution to science to extend our knowledge, though should be mindful they may not always be translatable into a bio-application as a pharmacological tool. I recommend minor corrections prior to acceptance.

I have the following specific feedback:

nAChR activity has been isolated for sea anemone peptides see Toxins 202214(10), 697; https://doi.org/10.3390/toxins14100697 modify discussion to reflect this work and include

Throughout the paper there are minor grammar changes to be fixed e.g., in the title “the” should be before sea anemone – this type of error is commonly found throughout the manuscript

Introduction – The ecology for sea anemones is incorrect in the first sentence. They are not predatory – they do not actively hunt their food but are opportunistic feeders. Additionally, their movement is not restricted but they do tend to settle in the one place giving the impression they are sessile, but they are not (coral is an example of sessile). If conditions are unfavourable, sea anemones will most definitely move.

The authors should cite primary research other than relying on reviews.

Author Response

This paper looks at the functional characterisation of some novel peptides with nAChR activity and a variation on the ICK fold. Function characterisation of any peptide is always important and a contribution to science to extend our knowledge, though should be mindful they may not always be translatable into a bio-application as a pharmacological tool. I recommend minor corrections prior to acceptance.

I have the following specific feedback:

nAChR activity has been isolated for sea anemone peptides see Toxins 202214(10), 697; https://doi.org/10.3390/toxins14100697 modify discussion to reflect this work and include

We added information about this work to the Discussion.

Throughout the paper there are minor grammar changes to be fixed e.g., in the title “the” should be before sea anemone – this type of error is commonly found throughout the manuscript

Corrected

Introduction – The ecology for sea anemones is incorrect in the first sentence. They are not predatory – they do not actively hunt their food but are opportunistic feeders. Additionally, their movement is not restricted but they do tend to settle in the one place giving the impression they are sessile, but they are not (coral is an example of sessile). If conditions are unfavourable, sea anemones will most definitely move.

The authors should cite primary research other than relying on reviews.

We modified the first sentence and add a reference. I would be happy to agree with you that" they do not actively hunt their food" but I saw it myself for sea anemones Urticina grebelnyi and Urticina eques. They actively move tentacles to the prey when it touches one of them. Zoologist consider sea anemones "opportunistic polyphagous predators (Quesada 2014)".

This paper looks at the functional characterisation of some novel peptides with nAChR activity and a variation on the ICK fold. Function characterisation of any peptide is always important and a contribution to science to extend our knowledge, though should be mindful they may not always be translatable into a bio-application as a pharmacological tool. I recommend minor corrections prior to acceptance.

I have the following specific feedback:

nAChR activity has been isolated for sea anemone peptides see Toxins 202214(10), 697; https://doi.org/10.3390/toxins14100697 modify discussion to reflect this work and include

We added information about this work to the Discussion.

Throughout the paper there are minor grammar changes to be fixed e.g., in the title “the” should be before sea anemone – this type of error is commonly found throughout the manuscript

Corrected

Introduction – The ecology for sea anemones is incorrect in the first sentence. They are not predatory – they do not actively hunt their food but are opportunistic feeders. Additionally, their movement is not restricted but they do tend to settle in the one place giving the impression they are sessile, but they are not (coral is an example of sessile). If conditions are unfavourable, sea anemones will most definitely move. The authors should cite primary research other than relying on reviews.

We modified the first sentence and add another reference. I would be happy to agree with you that "they do not actively hunt their food" but I saw it myself for sea anemones Urticina grebelnyi and Urticina eques. They actively move tentacles to the prey when it touches one of them. Zoologists consider sea anemones " opportunistic polyphagous predators (Quesada 2014)".

Reviewer 2 Report

This study isolates and characterises several novel peptides from sea anemone venom with moderate potency against several nicotinic acetylcholine receptors. Overall, the claims made in the manuscript are justified from the data. I have a few suggestions for the authors:

- Page 2 Line 54. This work hasn’t shown that the ICK fold, or modified ICK fold, is what “allows the most effective interaction with muscle and α9 neuronal nAChR subtypes” as written in the last line of the Introduction. This is hypothesised in the discussion but not shown via any data here. Please rephrase this to better reflect the work.

- Page 2 Line 58. Similar to above, the first line of the results is not actually shown “The venom of sea anemone Metridium senile showed significant anticholinergic activity.” The data in Fig 1 shows the venom prevents a-Bgt binding, not the block of Ach at synapses as would be imaged by using “anticholinergic” in the text.

- Figure 1. It would be easier to assess the data if individual data points are shown with the bar graph. In particular with the 0 mg/ml, the bar ends at 100%, but the error bars suggest there’s variability.

- Fig 2d. Are these samples a specific dilution of the HPLC trace? Or is the amount applied normalised to peak area? Unclear what the sample amounts are, even if it would be a rough estimate. It is also unclear which regions of the chromatogram in panel C relates to which fractions from panel D.

- Fig 3C. At a glance the new sequences look hardly similar to Pcf1a, AcaTx1, or PcTx1. If the comparison is going to be made, it could help to add a column after the sequences to indicate percent identity relative to Ms11a-1?

- Fig 4 is missing from the pdf.

- Table 1 and 2. It might be easier to present this data as micromolar, not nanomolar, as the values are quite large in nM.

- Page 6 Line 190. It would be good to clarify that this line is also in reference to different assays, not just species “A more significant species-selectivity was found for the fish and mammalian muscle receptor subtype (IC50 255 191 vs 1215 nM) (Table 2).

- Table 2 footnote b. For the not active samples, is this an n=2 or 3-6? This text is unclear “Ms11a-3 analyzed in 2 recordings on 3-6 oocytes.

- Fig 7. The disorder is quite obvious, but it could help readers by placing a label in the figure to show which loop is loop2.  

- Page 11 Line 293. This line isn’t the most accurate “Besides, an additional difference is the presence of a prolonged loop between the fifth and sixth cysteines in the novel anemone toxins Ms11a-1 − Ms11a-3, which is not present in other known spider ICK peptides.” Agreed that the loop in the peptides from this study are long (14 residues). But they are not that much longer than Tachystatin-B1 (11 residues; PDB 2DCV) or Hybrid atracotoxin (13 residues; PDB 2H1Z). I can understand highlighting the loop length, and its importance in nAChR activity, but it’s not that much longer in Ms11a1-3 than some other ICK peptides.

- Ref 56 is incorrectly cited for the TEVC.

Author Response

This study isolates and characterises several novel peptides from sea anemone venom with moderate potency against several nicotinic acetylcholine receptors. Overall, the claims made in the manuscript are justified from the data. I have a few suggestions for the authors:

- Page 2 Line 54. This work hasn’t shown that the ICK fold, or modified ICK fold, is what “allows the most effective interaction with muscle and α9 neuronal nAChR subtypes” as written in the last line of the Introduction. This is hypothesised in the discussion but not shown via any data here. Please rephrase this to better reflect the work.

We rephrased the sentence - " allows the effective interaction with the muscle and α9 nAChR subtypes ".

- Page 2 Line 58. Similar to above, the first line of the results is not actually shown “The venom of sea anemone Metridium senile showed significant anticholinergic activity.” The data in Fig 1 shows the venom prevents a-Bgt binding, not the block of Ach at synapses as would be imaged by using “anticholinergic” in the text.

Corrected. "The venom of sea anemone M. senile showed significant inhibition of labeled α-bungarotoxin binding to nAChR."

- Figure 1. It would be easier to assess the data if individual data points are shown with the bar graph. In particular with the 0 mg/ml, the bar ends at 100%, but the error bars suggest there’s variability.

We modified Figure 1

- Fig 2d. Are these samples a specific dilution of the HPLC trace? Or is the amount applied normalised to peak area? Unclear what the sample amounts are, even if it would be a rough estimate.

The samples were tested at concentrations corresponding to active concentrations of the venom. As shown in the figure legend, we separated 5.45 mg of venom and further used aliquots for testing to get final concentration 1 mg/ml if count amount of the venom. We modified the figure legend to make it clear.

It is also unclear which regions of the chromatogram in panel C relates to which fractions from panel D.

We corrected the figure legend to make it more clear.

- Fig 3C. At a glance the new sequences look hardly similar to Pcf1a, AcaTx1, or PcTx1. If the comparison is going to be made, it could help to add a column after the sequences to indicate percent identity relative to Ms11a-1?

Added to Fig 3C

- Fig 4 is missing from the pdf.

Corrected

- Table 1 and 2. It might be easier to present this data as micromolar, not nanomolar, as the values are quite large in nM.

Corrected

- Page 6 Line 190. It would be good to clarify that this line is also in reference to different assays, not just species “A more significant species-selectivity was found for the fish and mammalian muscle receptor subtype (IC50 255 191 vs 1215 nM) (Table 2).

The paragraph has been rewritten.

- Table 2 footnote b. For the not active samples, is this an n=2 or 3-6? This text is unclear “Ms11a-3 analyzed in 2 recordings on 3-6 oocytes.

Corrected

- Fig 7. The disorder is quite obvious, but it could help readers by placing a label in the figure to show which loop is loop2.  

Figure modified

- Page 11 Line 293. This line isn’t the most accurate “Besides, an additional difference is the presence of a prolonged loop between the fifth and sixth cysteines in the novel anemone toxins Ms11a-1 − Ms11a-3, which is not present in other known spider ICK peptides.” Agreed that the loop in the peptides from this study are long (14 residues). But they are not that much longer than Tachystatin-B1 (11 residues; PDB 2DCV) or Hybrid atracotoxin (13 residues; PDB 2H1Z). I can understand highlighting the loop length, and its importance in nAChR activity, but it’s not that much longer in Ms11a1-3 than some other ICK peptides.

We changed the sentence to make it more accurate. "Besides, an additional difference is the presence of a prolonged loop between the fifth and sixth cysteines in the novel anemone toxins Ms11a-1 − Ms11a-3, which is typically shorter in other known ICK peptides."

- Ref 56 is incorrectly cited for the TEVC.

References corrected

Reviewer 3 Report

Title: Replace “ICK” with “inhibitor cystine knot (ICK)”

Abstract

Line 6: Replace “systems and other” with “systems, and other”

Line 7: Replace “them considered” with “them are considered”

Line 9: Replace “nicotinic acetylcholine receptors” with “nAChRs”

Line 14: Replace “α9α10 nAChRs” with “heterologous rat α9α10 nAChR” and delete “neuronal”. The α7 subtype is also found in non-neuronal cells.

Line 17: Replace “have structures,” with “have strucutres”

Line 18: Replace “known” with “other known”

Line 19: Replace “Metridium senile” with “Metridium senile

Introduction

Line 34: Replace “cnidarians” with “cnidarian”

Line 42: Replace “analyses” with “analyses”

Line 44: Replace “, inhibitors” with “and, inhibitors”

Line 49: Replace “explored” with “exploited” and “the peptide toxins” with “peptide toxins”

Line 50-51: Consider deleting the statement “For the best of our knowledge, no peptides from sea anemones were characterized as modulators of nAChRs”. There is an article published on nAChR-acting toxins from the sea anemone Heteractis magnifica, DOI: 10.3390/toxins14100697.

Line 53: Replace “block” with “inhibit”

Line 54: Replace “ICK” with “inhibitor cystine knot (ICK)” and “interaction with muscle and α9” with “interaction with the muscle and α9α10”. Delete “neuronal”

Results

Line 58: Replace “Metridium senile” with “M. senile

Line 59: Replace “dose-dependently inhibited” with “dose-dependently and competitively inhibited”

Line 61: Delete “neuronal”

Line 63-64: It would be more appropriate to start from the lowest to highest venom concentrations. Replace “with the complete inhibition at a venom concentration of 1 mg/ml (97.2 ± 0.8 %) and 88.5 ± 1.9 % inhibition at 0.1 mg/ml (Fig. 1a).” with “than the human α7 nAChR, with 88.5 ± 1.9 % inhibition at 0.1 mg/ml and 97.2 ± 0.8 % at 1 mg/ml (Fig. 1a).” To say the venom completely inhibits the muscle nAChRs is not accurate.

Line 64: Replace “However, towards human α7 nAChR only 74.0 ± 1.2 % and 15 ± 8 % inhibition of specific [125I]α‐Bgt binding at a venom concentration of 1.0 and 0.1 mg/ml, respectively, was achieved (Fig. 1b)” with “For comparison, at the human α7 nAChR, 0.1 mg/ml and 1 mg/ml of venom inhibited 15 ± 8 % and 74.0 ± 1.2 %  of [125I]α‐Bgt binding, respectively (Fig. 1b).”

Line 66: Replace “so” with “Thus,”

Figure 1: Replace Y-axis label with “Specific binding (%)” and X-axis label with “Venom concentration (mg/ml)”. Change the order of the bars from lowest to highest concentration of venom.

Line 71: Insert “α1β1γδ” after “Torpedo californica

Line 74: Replace “stage” with “stage,”

Line 76: Replace “Torpedo nAChR” with “Toperdo α1β1γδ nAChRs” and replace “the gray shaded area” with “(gray shaded area)”

Line 79: Replace “components(Fig. 2C)” with “components (Fig. 2C)” and delete one of the “#”

Figure 2: For (a,c,e and f), ) replace X-axis label with “Time (min)”. For (b) replace Y-axis label with “Specific binding (%)” and X-axis label with “Concentration (mg/ml)”. For (d) replace Y-axis label with “Specific binding (%)”. Insert unit for the molecular mass in (c), (e) and (f).

Line 99: Replace “Torpedo” with “Torpedo

Line 106: Replace “On the base of obtained N-terminal sequences were designed primers” with “Based on the obtained N-terminal sequences, primers were designed”

Line 112: Replace “In case of” with “For”

Line 121: Replace “C-terminal” with “The C-terminal”

Line 122: Replace “post translationally” with “post-translationally”

Line 143: Delete “very”

Line 159: Replace “on their binding and blocking” with “for their binding and inhibitory”

Line 160: Replace “Ability” with “The ability”

Line 161: Delete “neuronal”

Line 163: Replace “, which collected in Table 1” with “(Table 1)”

Line 164: Replace “muscle-type” with “the muscle-type”

Line 166: Replace “maximal” with “most”

Table 1: Consider changing the title first row to “[125I] α-Bgt-competitive binding assay”. For the title second row, replace “…Torpedo californica nAChR” and “…human α7 nAChR” with “T. californica α1β1γδ nAChR” and “human α7 nAChR”, respectively.

Line 173-177: From Table 2, only the mouse muscle nAChR is listed, but the human muscle subtype is mentioned in line 175. Replace “For these reasons, we investigated the functional activity and selectivity towards different nAChR subtypes from some animal species only for peptide Ms11a-3. The ability of this peptide to block the acetylcholine-induced opening of the mouse and human muscle α1β1εδ receptor and the set of rat and human neuronal nAChR subtypes expressed in Xenopus oocytes, was studied (Fig. 5, Table 2).” with “For these reasons, we only investigated the functional activity and selectivity of peptide Ms11a-3 at acetylcholine (ACh)-evoked currents mediated by different nAChR subtypes of some animal species (mouse (m), rat (r) and human (h)) heterologously expressed in Xenopus laevis oocytes (Fig. 5, Table 2).”

Figure 5: Include representative traces for Ms11a-3 at one concentration, at the nAChRs tested. Replace Y-axis label with “Response (%)”.The muscle subtype is referred to as either α1β1εδ or α1β1δε in the manuscript. Please use one consistent label.  

Line 179: Replace “on most sensible” with “at most sensible”

Line 182: Replace “inserted” with “shown”

Line 183: Replace “The best blocking potency was found for the rat α9α10 nAChRs (IC50 = 202 nM). The second target was the mouse muscle α1β1εδ receptor with IC50 = 1215 nM.” with “Peptide Ms11a-3 was most potent at inhibiting rα9α10 nAChRs (IC50 = 202 nM), followed with mα1β1εδ (IC50 = 1215 nM).”

Line 184-185: Replace “efficacy” with “potency”

Line 185: Delete “neuronal”

Line 186-188: Consider deleting “The measured affinity for the human α7 nAChR in radioligand 2.2 times less than half inhibitory concentration in electrophysiological tests (IC50 19.8 vs 8.9 μM; see Tables 1 and 2) highlighting the difference between these approaches”. It is not sensible to compare the results from radioligand assay and electrophysiology experiments as the assay measures peptide binding whereas the latter measures functional responses.

Line 188: Replace “blocking potency of the” with “potency of Ms11a-3 at”

Line 189: Replace “receptor” with “receptors” and “IC50 8.9 vs 4.8 μM)” with “IC50 8.9 vs 4.8 μM, respectively)”

Line 190: Replace “apparently activity was less on human” with “the potency was less on the human”

Line 190-191: Consider deleting “A more significant species-selectivity was found for the fish and mammalian muscle receptor subtype (IC50 255 191 vs 1215 nM) (Table 2). See comments for line 186-188.

Line 194: Replace “oocites” with “oocytes”. Delete “in electrophysiology tests”

Table 2: Title: Replace “nAChR subtype” with “nAChR subtype a”, “IC50a” with “IC50” and “Hill slopea” with “Hill slope”

Line 196: Replace “the blocking percentage of nAChRs is not obvious” with “no inhibition observed”

Line 206: Replace “caused by” with “due to”

Figure 6: Disulfide bonds are indicated in figure as connected black lines. Please include in figure legend a description to indicate the disufilde bonds.

Discussion

Line 252: Replace “the predation” with “predation”

Line 257: Replace “were characterized” with “have been characterized”

Line 257-259: See comments for line 50-51

Line 261: Replace “blocking” with “inhibiting”

Line 265: Replace “Metridium” with “M.

Line 277: Replace “into the molecule of α-conotoxins” with “into Conus α-conotoxin peptides”

Line 281: Delete “neuronal”

Line 289: Replace “Inhibitor Cystine Knot” with “inhibitor cysteine knot”

Line 304, 306, 325 and 332: Replace “ICK-knot” with “ICK knot”

Line 318: Replace “the peptide” with “a peptide”

Line 321: Replace “Nav”with “voltage-gated sodium”

Line 347: Replace “Vetrebrata” with “vetrebrata”

Line 351: Replace “venom” with “the venom” and “evidently that” with “evidently,”

Line 354: Replace “M.senile” with “M. senile

Line 357: Delete “neuronal”

Conclusions

Line 362: Replace “Metridium” with “M.

Line 367: Replace “known” with “other known”

Materials and methods

Line 393: Replace “M. senile” with “M. senile

Line 411: Replace “E. coli” with “E. coli

Line 426: Replace “T. californica” with “T. californica

Line 441: Replace “Xenopus laevis” with “Xenopus laevis

Line 449: Replace “Two-electrode” with “The two-electrode”

Line 450: Replace “Xenopus laevis” with “X. laevis

Line 453: Replace “Acetylcholine” with “acetylcholine”

Line 440: Include statistical analysis used for the binding assay. See line 469-471

Line 455: Justify the use of these ACh concentrations. Are they the ACh EC50 of each subtype?

Line 465, 467 and 468: Replace “IC50” with “IC50

Line 470: Replace “Mean ± Standard” with “mean ± standard”

Supplementary information

Table S1: Title: Replace “Mw” with “MW”

Author Response

Dear Reviewer,

We are thankful for your valuable comments. Below is the point-by-point discussion of the issues you raised.

Title: Replace “ICK” with “inhibitor cystine knot (ICK)

Corrected

Abstract

Done Line 6: Replace “systems and other” with “systems, and other”

Corrected

Done Line 7: Replace “them considered” with “them are considered”

Corrected

Done Line 9: Replace “nicotinic acetylcholine receptors” with “nAChRs”

Corrected

Done Line 14: Replace “α9α10 nAChRs” with “heterologous rat α9α10 nAChR” and delete “neuronal”. The α7 subtype is also found in non-neuronal cells –

Corrected

Done Line 17: Replace “have structures,” with “have strucutres”

Corrected

Done Line 18: Replace “known” with “other known”

Corrected

Done Line 19: Replace “Metridium senile” with “Metridium senile

 Corrected

Introduction

Done Line 34: Replace “cnidarians” with “cnidarian”

Corrected

Done Line 42: Replace “analyses” with “analyses”

Corrected

Done Line 44: Replace “, inhibitors” with “and, inhibitors”

Corrected

Done Line 49: Replace “explored” with “exploited” and “the peptide toxins” with “peptide toxins”

Corrected

Line 50-51: Consider deleting the statement “For the best of our knowledge, no peptides from sea anemones were characterized as modulators of nAChRs”. There is an article published on nAChR-acting toxins from the sea anemone Heteractis magnifica, DOI: 10.3390/toxins14100697. –

We deleted this statement from introduction and added information about this work to the Discussion.

Done Line 53: Replace “block” with “inhibit”

Corrected

Done Line 54: Replace “ICK” with “inhibitor cystine knot (ICK)” and “interaction with muscle and α9” with “interaction with the muscle and α9α10”. Delete “neuronal”

 Corrected

Results

Done Line 58: Replace “Metridium senile” with “M. senile

Corrected

Done Line 59: Replace “dose-dependently inhibited” with “dose-dependently and competitively inhibited”

Corrected

Done Line 61: Delete “neuronal”

Corrected

Done Line 63-64: It would be more appropriate to start from the lowest to highest venom concentrations. Replace “with the complete inhibition at a venom concentration of 1 mg/ml (97.2 ± 0.8 %) and 88.5 ± 1.9 % inhibition at 0.1 mg/ml (Fig. 1a).” with “than the human α7 nAChR, with 88.5 ± 1.9 % inhibition at 0.1 mg/ml and 97.2 ± 0.8 % at 1 mg/ml (Fig. 1a).” To say the venom completely inhibits the muscle nAChRs is not accurate.

Corrected

Done Line 64: Replace “However, towards human α7 nAChR only 74.0 ± 1.2 % and 15 ± 8 % inhibition of specific [125I]α‐Bgt binding at a venom concentration of 1.0 and 0.1 mg/ml, respectively, was achieved (Fig. 1b)” with “For comparison, at the human α7 nAChR, 0.1 mg/ml and 1 mg/ml of venom inhibited 15 ± 8 % and 74.0 ± 1.2 %  of [125I]α‐Bgt binding, respectively (Fig. 1b).”

Corrected

Done Line 66: Replace “so” with “Thus,”

Corrected

Figure 1: Replace Y-axis label with “Specific binding (%)” and X-axis label with “Venom concentration (mg/ml)”. Change the order of the bars from lowest to highest concentration of venom.

Figure modified

Done Line 71: Insert “α1β1γδ” after “Torpedo californica

Corrected

Done Line 74: Replace “stage” with “stage,”

Corrected

Done Line 76: Replace “Torpedo nAChR” with “Toperdo α1β1γδ nAChRs” and replace “the gray shaded area” with “(gray shaded area)”

Corrected

Done Line 79: Replace “components(Fig. 2C)” with “components (Fig. 2C)” and delete one of the “#”

Corrected

Figure 2: For (a,c,e and f), ) replace X-axis label with “Time (min)”. For (b) replace Y-axis label with “Specific binding (%)” and X-axis label with “Concentration (mg/ml)”. For (d) replace Y-axis label with “Specific binding (%)”. Insert unit for the molecular mass in (c), (e) and (f).

Figure modified

Done Line 99: Replace “Torpedo” with “Torpedo

Corrected

Done Line 106: Replace “On the base of obtained N-terminal sequences were designed primers” with “Based on the obtained N-terminal sequences, primers were designed”

Corrected

Done Line 112: Replace “In case of” with “For”

Corrected

Done Line 121: Replace “C-terminal” with “The C-terminal”

Corrected

Done Line 122: Replace “post translationally” with “post-translationally”

Corrected

Done Line 143: Delete “very”

Corrected

Done Line 159: Replace “on their binding and blocking” with “for their binding and inhibitory”

Corrected

Done Line 160: Replace “Ability” with “The ability”

Corrected

Done Line 161: Delete “neuronal”

Corrected

Done Line 163: Replace “, which collected in Table 1” with “(Table 1)”

Corrected

Done Line 164: Replace “muscle-type” with “the muscle-type”

Corrected

Done Line 166: Replace “maximal” with “most”

Corrected

Table 1: Consider changing the title first row to “[125I] α-Bgt-competitive binding assay”. For the title second row, replace “…Torpedo californica nAChR” and “…human α7 nAChR” with “T. californica α1β1γδ nAChR” and “human α7 nAChR”, respectively.

Corrected

Line 173-177: From Table 2, only the mouse muscle nAChR is listed, but the human muscle subtype is mentioned in line 175. Replace “For these reasons, we investigated the functional activity and selectivity towards different nAChR subtypes from some animal species only for peptide Ms11a-3. The ability of this peptide to block the acetylcholine-induced opening of the mouse and human muscle α1β1εδ receptor and the set of rat and human neuronal nAChR subtypes expressed in Xenopus oocytes, was studied (Fig. 5, Table 2).” with “For these reasons, we only investigated the functional activity and selectivity of peptide Ms11a-3 at acetylcholine (ACh)-evoked currents mediated by different nAChR subtypes of some animal species (mouse (m), rat (r) and human (h)) heterologously expressed in Xenopus laevis oocytes (Fig. 5, Table 2).”

Corrected

Figure 5: Include representative traces for Ms11a-3 at one concentration, at the nAChRs tested. Replace Y-axis label with “Response (%)”.The muscle subtype is referred to as either α1β1εδ or α1β1δε in the manuscript. Please use one consistent label.  

Figure modified

Done Line 179: Replace “on most sensible” with “at most sensible”

Corrected

Done Line 182: Replace “inserted” with “shown”

Corrected

Done Line 183: Replace “The best blocking potency was found for the rat α9α10 nAChRs (IC50 = 202 nM). The second target was the mouse muscle α1β1εδ receptor with IC50 = 1215 nM.” with “Peptide Ms11a-3 was most potent at inhibiting rα9α10 nAChRs (IC50 = 202 nM), followed with mα1β1εδ (IC50 = 1215 nM).”

Corrected

Done Line 184-185: Replace “efficacy” with “potency”

Corrected

Done Line 185: Delete “neuronal”

Corrected

Line 186-188: Consider deleting “The measured affinity for the human α7 nAChR in radioligand 2.2 times less than half inhibitory concentration in electrophysiological tests (IC50 19.8 vs 8.9 μM; see Tables 1 and 2) highlighting the difference between these approaches”. It is not sensible to compare the results from radioligand assay and electrophysiology experiments as the assay measures peptide binding whereas the latter measures functional responses.

The paragraph has been rewritten.

Done Line 188: Replace “blocking potency of the” with “potency of Ms11a-3 at”

Corrected

Done Line 189: Replace “receptor” with “receptors” and “IC50 8.9 vs 4.8 μM)” with “IC50 8.9 vs 4.8 μM, respectively)”

Corrected

Done Line 190: Replace “apparently activity was less on human” with “the potency was less on the human”

Corrected

Line 190-191: Consider deleting “A more significant species-selectivity was found for the fish and mammalian muscle receptor subtype (IC50 255 191 vs 1215 nM) (Table 2). See comments for line 186-188.

The paragraph has been rewritten.

Done Line 194: Replace “oocites” with “oocytes”. Delete “in electrophysiology tests”

Corrected

Table 2: Title: Replace “nAChR subtype” with “nAChR subtype a”, “IC50a” with “IC50” and “Hill slopea” with “Hill slope”

Corrected

Line 196: Replace “the blocking percentage of nAChRs is not obvious” with “no inhibition observed”

Corrected

Done Line 206: Replace “caused by” with “due to”

Corrected

Figure 6: Disulfide bonds are indicated in figure as connected black lines. Please include in figure legend a description to indicate the disufilde bonds.

 Added to figure legend.

Discussion

Done Line 252: Replace “the predation” with “predation”

Corrected

Done Line 257: Replace “were characterized” with “have been characterized”

Corrected

Line 257-259: See comments for line 50-51

corrected

Done Line 261: Replace “blocking” with “inhibiting”

Corrected

Done Line 265: Replace “Metridium” with “M.

Corrected

Done Line 277: Replace “into the molecule of α-conotoxins” with “into Conus α-conotoxin peptides” Corrected to " into α-conotoxin peptides"

Done Line 281: Delete “neuronal”

Corrected

Done Line 289: Replace “Inhibitor Cystine Knot” with “inhibitor cysteine knot”

Corrected

Done Line 304, 306, 325 and 332: Replace “ICK-knot” with “ICK knot”

Corrected to ICK fold

Done Line 318: Replace “the peptide” with “a peptide”

Corrected

Done Line 321: Replace “Nav”with “voltage-gated sodium”

Corrected

Done Line 347: Replace “Vetrebrata” with “vetrebrata”

Corrected

Done Line 351: Replace “venom” with “the venom” and “evidently that” with “evidently,”

Corrected

Done Line 354: Replace “M.senile” with “M. senile

Corrected

Done Line 357: Delete “neuronal”

Corrected

Conclusions

Done Line 362: Replace “Metridium” with “M.

Corrected

Done Line 367: Replace “known” with “other known”

Corrected

Materials and methods

Done Line 393: Replace “M. senile” with “M. senile

Corrected

Done Line 411: Replace “E. coli” with “E. coli

Corrected

Done Line 426: Replace “T. californica” with “T. californica

Corrected

Done Line 441: Replace “Xenopus laevis” with “Xenopus laevis

Corrected

Done Line 449: Replace “Two-electrode” with “The two-electrode”

Corrected

Done Line 450: Replace “Xenopus laevis” with “X. laevis

Corrected

Done Line 453: Replace “Acetylcholine” with “acetylcholine”

Corrected

Line 440: Include statistical analysis used for the binding assay. See line 469-471

Added to the methods

Line 455: Justify the use of these ACh concentrations. Are they the ACh EC50 of each subtype?

The concentration of ACh selected in this experiment to induce the opening of nAChR channels was based on the EC50 values of ACh for different nAChR subtypes. At this concentration, ACh can induce channel opening without causing channel desensitization. The concentration of ACh is widely used in other studies.

Done Line 465, 467 and 468: Replace “IC50” with “IC50

Corrected

Done Line 470: Replace “Mean ± Standard” with “mean ± standard”

Corrected

Supplementary information

Table S1: Title: Replace “Mw” with “MW”

Corrected

Reviewer 4 Report

This manuscript investigates novel peptides from the sea anemone Metridium senile. 

Other anemones are well known to produce peptides that modulate sodium and potassium channels as well as ASIC and TRP channels.  The authors found that crude venom of Metridium blocks various nicotinic receptors (nAChRs) and set about purifying and characterizing the responsible peptides. Since sea anemones are not well know to produce neurotoxins with this pharmacological activity, this manuscript is of intrinsic interest.

The authors went on to isolate four novel peptides, cloned the genes, expressed the proteins in E. coli, and showed their efficacy against various nAChR subtypes using oocyte electrophysiology.  They cite data not included in this manuscript that these peptides do not affect other channels including ASIC and TRP channels. They also obtained structural information on two peptides using NMR and worked out the disulfide bond arrangements to find that these form a novel Inhibitor Cysteine Knot (ICK) fold.

The manuscript is fairly well written (although see some suggestions below).  Also, the authors have previously deposited the peptide structures and sequences in the PDB database and in PubMed, and those entries should be supported by a published manuscript.   The authors claim that these novel peptides could represent new lead compounds for nAChR inhibitors, and this reviewer agrees.

Two minor issues: 

On line 50 and again on line 257, the authors claim that venom from no other sea anemone has shown effects on nicotinic receptors.  However, a very recent report in this journal (Toxins 2022, 14, 697. https://doi.org/10.3390/toxins14100697) makes a similar claim, although this earlier report shows that the four peptides found in the venom of the sea anemone Heteractis magnifica (completely different sequences from M. senile) modulate nAChRs (in one case, enhancing alpha7 receptor responses).  Furthermore, that publication reports that the Heteractis peptides also block rat ASIC channels, unlike the Metridium peptides. The authors should cite this paper (which probably appeared as they were finishing their manuscript for submission).  Further, they may consider including the ASIC data in the body of their report to make the claim that the Metridium peptides are the first sea anemone nAChR antagonists that do not also affect ASIC channels.

Although the manuscript is well written, there are a few English usage issues. For instance, on line 24 the authors state that anemones “have no vision and a centralized or coordinated nervous system” which sounds like they think anemones do have a centralized nervous system, and I don’t think they want to say that.  Better constructs might be “have no vision nor a …” or “have no vision and no centralized….”

On line 106, it’s not clear what the authors are trying to say.  I think they are saying “DNA primers for RACE (rapid amplification of cDNA ends) were designed based on N-terminal peptide amino acid sequences and used M. senile total RNA as a PCR template.”

Also, on line 50, the expression normally is “To the best of our knowledge”, not “For…”.  But these are very minor, and line 50 needs re-writing anyway.

Finally, this reviewer has no competence to judge whether the ICK folds found in this manuscript are indeed unique.  Hopefully, another reviewer will address this issue.

Author Response

Dear Reviewer,

We are thankful for your valuable comments. Below is the point-by-point discussion of the issues you raised.

This manuscript investigates novel peptides from the sea anemone Metridium senile

Other anemones are well known to produce peptides that modulate sodium and potassium channels as well as ASIC and TRP channels.  The authors found that crude venom of Metridium blocks various nicotinic receptors (nAChRs) and set about purifying and characterizing the responsible peptides. Since sea anemones are not well know to produce neurotoxins with this pharmacological activity, this manuscript is of intrinsic interest.

The authors went on to isolate four novel peptides, cloned the genes, expressed the proteins in E. coli, and showed their efficacy against various nAChR subtypes using oocyte electrophysiology.  They cite data not included in this manuscript that these peptides do not affect other channels including ASIC and TRP channels. They also obtained structural information on two peptides using NMR and worked out the disulfide bond arrangements to find that these form a novel Inhibitor Cysteine Knot (ICK) fold.

The manuscript is fairly well written (although see some suggestions below).  Also, the authors have previously deposited the peptide structures and sequences in the PDB database and in PubMed, and those entries should be supported by a published manuscript.   The authors claim that these novel peptides could represent new lead compounds for nAChR inhibitors, and this reviewer agrees.

Two minor issues: 

On line 50 and again on line 257, the authors claim that venom from no other sea anemone has shown effects on nicotinic receptors.  However, a very recent report in this journal (Toxins 2022, 14, 697. https://doi.org/10.3390/toxins14100697) makes a similar claim, although this earlier report shows that the four peptides found in the venom of the sea anemone Heteractis magnifica (completely different sequences from M. senile) modulate nAChRs (in one case, enhancing alpha7 receptor responses).  Furthermore, that publication reports that the Heteractis peptides also block rat ASIC channels, unlike the Metridium peptides. The authors should cite this paper (which probably appeared as they were finishing their manuscript for submission). Further, they may consider including the ASIC data in the body of their report to make the claim that the Metridium peptides are the first sea anemone nAChR antagonists that do not also affect ASIC channels.

We added this information to the Discussion

Although the manuscript is well written, there are a few English usage issues. For instance, on line 24 the authors state that anemones “have no vision and a centralized or coordinated nervous system” which sounds like they think anemones do have a centralized nervous system, and I don’t think they want to say that.  Better constructs might be “have no vision nor a …” or “have no vision and no centralized….”

On line 106, it’s not clear what the authors are trying to say.  I think they are saying “DNA primers for RACE (rapid amplification of cDNA ends) were designed based on N-terminal peptide amino acid sequences and used M. senile total RNA as a PCR template.”

Corrected

Also, on line 50, the expression normally is “To the best of our knowledge”, not “For…”.  But these are very minor, and line 50 needs re-writing anyway.

Line 50 deleted

Finally, this reviewer has no competence to judge whether the ICK folds found in this manuscript are indeed unique.  Hopefully, another reviewer will address this issue.